# Clusters, Assemblies and Aggregates of Tumor Cells in the Blood of Breast Cancer Patients; Composition, Mode of Action, Detection and Impact on Metastasis and Survival

**Urszula Smietanka, Małgorzata Szostakowska-Rodzos, Sylwia Tabor, Anna Fabisiewicz** and **Ewa A. Grzybowska \***

Molecular and Translational Oncology, Maria Sklodowska-Curie National Research Institute of Oncology, Roentgena 5, 02-781 Warsaw, Poland; urszula.smietanka@pib-nio.pl (U.S.); malgorzata.szostakowska@pib-nio.pl (M.S.-R.); Sylwia.Tabor@pib-nio.pl (S.T.); anna.fabisiewicz@pib-nio.pl (A.F.)
\* Correspondence: ewa.grzybowska@pib-nio.pl

**Abstract:** Circulating tumor cells (CTCs) are gaining momentum as a diagnostic tool and therapeutic target. CTC clusters are more metastatic, but harder to study and characterize, because they are rare and the methods of isolation are mostly focused on single CTCs. This review highlights the recent advances to our understanding of tumor cell clusters with the emphasis on their composition, origin, biology, methods of detection, and impact on metastasis and survival. New approaches to therapy, based on cluster characteristics are also described.

**Keywords:** circulating tumor cells; clusters; breast cancer; metastasis

## 1. Introduction

Breast cancer is the most common invasive cancer in women and remains the second leading cause of cancer-related death in women after lung cancer. Significant improvement in early diagnostic in recent years followed by new methods of treatment led to a reduction in mortality but still metastatic breast cancer (MBC) remains incurable. This is mainly due to morphological and molecular heterogeneity between tumors as well as within a single tumor, which may be responsible for acquisition of the resistance to treatment.

Breast cancers are subdivided into subtypes based on the gene-expression profiling according to concurrent expression of the three types of receptors: estrogen and progesterone hormone receptors (HR) and HER2 (human epidermal growth factor receptor) [1,2]. These subtypes show different clinical behaviors and their characterization makes significant difference in the treatment approach. Luminal tumors (hormone-dependent) comprise subtype A (high HR expression, low Ki-67 proliferation index, HER2-, good prognosis) and subtype B (lower HR expression, high Ki-67, HER2+/−, worse prognosis), both may be treated with hormonal therapies. The other subtypes include HER2+, HR- (respond to HER2-targeted therapy), and triple-negative breast cancer (TNBC), without receptor expression and the most aggressive, treated with chemotherapy.

Despite relatively good prognosis, relapse due to metastasis in luminal breast cancer is larger in numbers than in other subtypes. It is because this subtype constitutes more than 70% of the cases. Thus, it is of utmost importance to correctly classify luminal tumors into those for which the probability of metastasis is high and those which are unlikely-to-metastasize, since it would influence the treatment decisions [3].

The presumptive precursors of distant metastases are circulating tumor cells (CTCs). CTCs have been detected in blood and described as early as in 1896 by Thomas Ashworth [4]. CTCs are defined as circulating cells in the peripheral blood whose antigenic or genetic characteristics correspond to those of a particular type of tumor. CTCs represent tumor cells that escaped from the tumor mass and translocated into the bloodstream/lymphatic system; this translocation can be passive (tumor cell shedding into leaky

vessels) or active, in which cells acquire invasive properties and degrade extracellular matrix, clearing the way for intravasation. Either way, these cells have a potential to extravasate and give rise to metastases, albeit not efficiently.

In the blood of patients, CTCs are very rare, occurring in the amount of 1 cell in $10^5$–$10^7$ mononuclear cells. In comparison to the classical biopsy of metastatic tissue, the isolation of CTCs is more advantageous in several respects: blood sampling is easier, cheaper, less invasive for the patient, and can be safely repeated while tracing the neoplastic process. Recently circulating tumor cells (CTCs) enumeration and characterization gained recognition as a reliable tool to stratify patients and choose treatment approach. CTC are also supposed to serve as a prognostic marker to monitor efficacy of adjuvant therapy and for early detection of minimal residual disease [5,6].

CTCs may be used to monitor oncogenic changes in disseminating cancers [7,8]. Analyzing the molecular profile of CTCs can therefore facilitate development of personalized treatment for each patient, which will avoid unnecessary or ineffective treatment [5,9]. It seems that in the future analysis of cancer cells in the collected peripheral blood (liquid biopsy) will be included into screening diagnostics, which would enable the detection and monitoring of cancer on the basis of a blood test.

CTCs are usually single, but sometimes they are associated in multicellular groupings (clusters), homo- or heterotypic. It was demonstrated that CTC clusters are much more metastatic than single cells. Different types of CTC clusters, their features, detection methods, and their impact on metastasis and treatment is described in this review.

## 2. Types of CTC Clusters and Cellular Assemblies Present in the Circulation

### 2.1. CTC Clusters

CTCs have been shown to occur in a form of homo- and heterotypic clusters, which are associated with enhanced metastatic potential, compared to single CTCs (Figure 1). CTC clusters are rare, constituting about 7.6% of CTCs. They are defined as nucleated, multicellular ($\geq$3 cells) entities expressing cytokeratins (CK+) and often, but not necessarily, EpCAM (epithelial cell adhesion molecule) and not expressing blood cells markers (CD45-). Homotypic clusters comprise only tumor cells and preserve epithelial cell–cell junctions. Heterotypic clusters comprise tumor cells, but also tumor-associated populations of immune cells and stromal cells (fibroblasts). These associated cells do not have carcinogenic mutations, but their phenotype is changed and serves to promote tumor progression. Macrophages and neutrophils undergo polarization to anti-tumoral and pro-tumoral populations, and pro-tumoral population can associate with CTCs [10,11]. Cancer-associated fibroblasts (CAFs) support tumor growth by extracellular matrix (ECM) re-modeling and promoting angiogenesis. CAFs are perpetually activated and express myofibroblastic markers, in particular $\alpha$-smooth muscle actin. They also have been shown to contribute to immune escape mediated by secretion of cytokines, chemokines, and the crosstalk with immune cells [12]. CAFs are also engaged in the crosstalk with tumor cells, resulting in a deposition of fibrillar ECM proteins like collagens, fibronectin, and laminins and increased ECM stiffness. In effect, CAFs create pro-tumoral microenvironment and enhance tumor progression in situ. Duda et al. [13], have shown that activated fibroblasts are not only a part of the pro-tumorigenic reactive stroma, but also form heterotypic CTC clusters which enhance metastasis by promoting the survival of tumor cells in the circulation and at the metastatic site. Brief schematic of possible types of clusters is presented in Figure 2 and a description of their specific markers in Table 1.

Another component of CTC clusters are platelets. Platelets, non-nucleated cell elements generated by fragmentation of megakaryocytes in bone marrow, are engaged in blood clothing and immune response. They were shown to surround and cloak CTCs [14], so platelet-covered CTCs are sometimes called "microemboli." Platelet cloaking may confer some protection against shear stress in the circulation and induce changes in tumor cells metabolism toward glycolysis [15]. Akolkar et al. [16] introduced a term "circulating en-

sembles of tumor-associated cells" (C-ETACs), describing heterotypic ensembles of tumor emboli, immune cells, and fibroblasts, so, in an essence, a form of heterotypic CTC clusters.

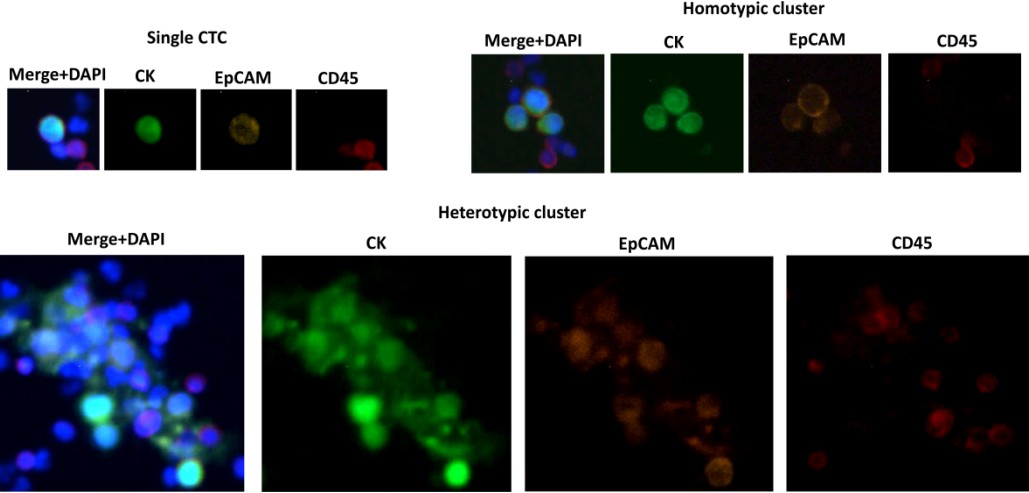

**Figure 1.** Examples of single CTC and homotypic and heterotypic CTC clusters. Staining: tumor markers (pan-cytokeratin—CK, EpCAM), WBC (CD45), nuclei (DAPI). Luminal breast cancer, image generated using CytoTrack system (M. Szostakowska-Rodzos).

## Types of CTC assemblies

**Figure 2.** Types of CTC clusters present in the circulation. Homotypic clusters comprise tumor cells, while heterotypic comprise also stromal cells (fibroblasts) and immune cells, with a special role of neutrophils. M2 macrophages can fuse with CTC producing large hybrid cells expressing markers and surface receptors characteristic to both types of cells.

**Table 1.** Specific markers used in detection of CTCs, CTC clusters and other circulating, cancer-associated cells. +: expression, −: no expression, +/−: expression in some cells.

| Description | Markers |
| --- | --- |
| CTC | CK+ (cytokeratins 8+, 18+, 19+), CD45−, EpCAM+, DAPI+ |
| CTC-neutrophil cluster | CK+/− (cytokeratins 8+, 18+, 19+), CD45+/−, EpCAM+/−, DAPI+, MPO (neutrophil myeloperoxidase)+/− |
| CTC-macrophage fusion | CK+ (cytokeratins 8+, 18+, 19+), CD45+, EpCAM+, DAPI+, CD14+ |
| NET | MPO (neutrophil myeloperoxidase)+, H3Cit (citrullinated histone H3)+, DAPI+ |

CTCs were also shown to associate with activated, circulating endothelial cells (CECs) [17] which represent yet another form of heterotypic CTC clusters and were suggested to protect CTCs from anoikis.

### 2.2. CTC-Neutrophil Clusters

White blood cells (WBC) include lymphocytes, monocytes, and granulocytes. Neutrophils, the most abundant granulocytes, constitute the first response in host defense against bacterial infection. Their role in cancer is complex; it is known for some time that neutrophil population can be polarized into anti-tumoral N1 phenotype, which promote tumor cell clearance and pro-tumoral tumor-associated neutrophils (TANs) with N2 phenotype, which have been shown to promote tumorigenesis [11]. About 3.4% of CTCs found in the bloodstream of breast cancer patients were shown to associate with neutrophils and this association was linked to the worse progression-free survival [18].

CTC-neutrophil clusters are formed via cell–cell adhesion, in particular with vascular cell adhesion molecule 1 (VCAM-1), involved with trans-endothelial cell migration. Thus, association with neutrophils may facilitate intravasation [19].

Szczerba et al. [18] have shown in the mouse model that CTC-neutrophil clusters promote tumor cell proliferation, cell cycle progression, and metastatic seeding. Single-cell RNA-seq analysis of CTC-WBC clusters vs. single CTCs indicated that the IL6 and IL1β signaling pathways are essential in the CTC-neutrophil crosstalk that leads to increased proliferation of cancer cells. The authors also identified several mutations present in these clusters, including the most important mutation in TLE1.

### 2.3. Neutrophil Extracellular Traps (NETs)

Neutrophils also contribute to metastasis via NETs formation. The phenomenon of NET was described in 2004 by Brinkmann et al. [20] as yet another form of innate response designed to entrap and kill bacteria. However, later on it was shown that NETs are also responsible for trapping CTCs and enhancing metastasis [21]. Neutrophils were shown to decondense their chromatin and extrude DNA in a form of fibrils decorated with antimicrobial peptides and enzymes, including neutrophil elastase, cathepsin G, and myeloperoxidase (MPO). NETs are degraded mostly by nucleases but not cleared by phagocytes, so their degradation takes longer. NETosis is triggered by microbe-associated factors, like bacterial LPS, but also by many other endogenous stimuli, including cytokines, platelets, activated endothelial cells, and some metabolites [22–24]. NETs have been shown to sequester CTCs via interactions with β1-integrin [25]. Trapped CTCs are more prone to adhesion to capillaries and subsequent extravasation to target organs [21]. Using intravital imaging Park et al. demonstrated in the mouse model that cancer cells induce metastasis-supporting neutrophil extracellular DNA traps [26].

### 2.4. Macrophage-Tumor Cell Fusion

As with neutrophils, macrophages can have pro- or anti-tumorigenic properties. Tumor microenvironment (TME)-infiltrating macrophages undergo M1 or M2 polarization,

which plays a role in cancer progression. M1 phenotype, triggered by bacterial lipopolysaccharide (LPS) or cytokines is associated with anti-tumor properties, while M2 phenotype is pro-tumorigenic, enhancing proliferation, survival, invasiveness, and immunosuppression [10,27]. M2 phenotypes have been observed to fuse with tumor cells, creating a large ($\geq$30 μm diameter), polymorphic, mono- or polynuclear hybrid cells, with dual, epithelial, and myeloid features. These hybrids expressing together CK+/EpCAM+ and CD14/CD45+, are found in the blood of patients with many cancer types, including breast cancer patients [28–31]. In these fused cells cytokeratin pattern is diffused, resembling mesenchymal cells, not filamentous, as in epithelial cells [32]. Interestingly, fusion can be partial, and occurs via tunneling nanotubes (TNTs), cellular projections that allow physical connection between the cytoplasm of both cells and transport of different cargos [33]. Permanent cell fusion produces a true hybrid cell with enhanced metastatic phenotype and surface receptors of both types.

Fusion is generally considered to confer migratory and invasive properties, enhancing metastasis. This was demonstrated by the study of the artificial fusion between breast tumor cells and macrophages; the fusion was shown to activate the Wnt/β-catenin signaling pathway via TCF/LEF transcription factor activating downstream targets [34]. Fused cells have been also described as displaying enhanced expression of the mesenchymal markers. Moreover, fusion of tumor cells with macrophages may lead to chemoresistance and immunosuppression [27].

## 3. The Origin of CTC Clusters

It has been demonstrated in an elegant mouse model experiment that CTC clusters are poly/oligoclonal, so they do not derive from a single, proliferating cell [35,36]. However, there are still two controversies surrounding the origin of CTC clusters: (1) do they exit primary tumor already as a cluster or do they form later as a result of aggregation and (2) do they derive from the edge or from the center of primary tumor?

### 3.1. Clusters Shedding vs. Intravascular Aggregation

In a seminal paper by Aceto et al. [35] the authors argue that CTC clusters cannot arise from intravascular aggregation, because in their mouse model experiments differentially stained tumor cells do not show any mixed seeding. Indeed, the circulation seems to be not the best place to aggregate, since it provides very hostile conditions for epithelial cells: shear stress, oxidative stress, and immune attack. Additionally, it should be mentioned that CTCs lifespan in the circulation is rather short—hours or minutes, even less in case of CTC clusters, giving no time for aggregation events. Despite this, using intravital microscopic imaging Liu et al. [37], demonstrated aggregation of individual tumor cells resulting in cluster formation, which were not generated by collective migration and cohesive shedding. The authors have shown that these clusters were enriched in breast cancer stem cell marker CD44, resulting in homophilic interactions and multicellular aggregation, which further initiates CD44-PAK2 interaction, resulting in activation of FAK signaling [37]. Observed clusters were shown to be linked to polyclonal metastasis, providing another explanation for this phenomenon.

### 3.2. Edge vs. Center

While it seems logical that molecularly diverse and more oxygenated tumor edge can shed CTC clusters more easily than hypoxic and partially necrotic inner core, there are evidences that CTC clusters in breast cancer derive from the hypoxic regions of the primary tumor and that these regions retain functional blood vessels [38]. Moreover, it was demonstrated that hypoxia results in cell–cell junction upregulation and intravasation of CTC clusters [38]. This association has the unexpected implications for breast cancer therapy; in the same paper it was demonstrated that targeting vascular endothelial growth factor A (VEGFA) in order to reduce primary tumor size is effective in itself, but increases CTC clusters and metastasis, via reducing angiogenesis and increasing hypoxia.

## 4. Markers and Phenotypes of CTC Clusters

### 4.1. Stemness

It is well-known that cancer cells may possess the features characteristic for stem cells, including cell renewal and formation of heterogeneous cell lineages [39]. Therefore, cells with these properties are believed to be critical in treatment failure and metastasis formation. Currently, cancer stem cells (CSCs) are one of the main focuses in cancer research. There are several markers used for CSCs identification depending on the cancer type (reviewed: [40]). Of all markers the combination of CD44, CD24, CD133, and ALDH-1 seems to be the most universal for CSCs identification in solid tumors. As indicated previously, newest research highlighted that CD44 might be critical for CTCs cluster formation via its target PAK2 [37,41], leading to multicellular aggregation. Moreover, in vitro studies on colon cancer cell line HCT-116 highlighted that post-sphere cultured cells that formed clusters had higher expression of stem markers than single cells [42]. This suggests that stemness might play a role in cluster formation. Stem cell properties are associated with expression of transcription factors characteristic for embryonic cells—SOX2, NANOG, OCT4. However, in healthy differentiated cells expression of these transcription factors is silenced. This inhibition of expression occurs at early stages of differentiation and should be permanent for fully differentiated cells. Recent studies indicated promoter methylation as possible regulatory system for long-term stem-related genes inhibition in tracheal cells [43]. Therefore, it is probable that changes in methylation status of stem-associated transcription factors are crucial for CSCs determination. Accordingly, major differences in methylation patterns between single CTCs and clustering CTCs were reported: clustering CTCs were characterized by hypomethylation of binding sites for major stemness regulators—OCT4, NANOG, SOX2, and SIN3A. Interestingly this methylation pattern was found to be exclusive for CTCs clusters, as dissociation reverts the methylation profile of CTC clusters and suppresses metastasis [44]. Many studies suggest that stem cell properties and cluster formation are linked. However, big cohort studies assessing clinical significance of stemness in CTC clusters are still to come.

### 4.2. Epithelial-Mesenchymal Plasticity

Epithelial-mesenchymal transition (EMT) is commonly associated with more invasive carcinoma; however, it seems that the highest tumorigenicity and metastatic potential can be attributed to the hybrid epithelial-mesenchymal (E/M) phenotype, in which markers of both types are overexpressed [45–47]. E/M phenotype is not some transitional state occurring during EMT, it is a stable, long-lasting phenotype. Hybrid phenotype is particularly eminent in CTCs, which display a wide spectrum of epithelial-mesenchymal plasticity (EMP) (reviewed in [48,49]). For example, it was demonstrated that more than 75% of CTCs from patients with metastatic breast cancer co-express epithelial cytokeratins with vimentin, and *N*-cadherin [50]. Hybrid phenotype traits are associated with increased migration, invasion, and survival, hence, these cells are considered more metastatic.

Some confusion pertains to epithelial cell marker E-cadherin. Long regarded as tumor suppressor, its expression was associated with good prognosis and its loss during EMT with increased invasiveness. However, recent findings undermine this view, showing that E-cadherin expression, while indeed limiting invasiveness (meant as the ability to degrade extracellular matrix), is in fact a survival factor, required for metastasis in multiple models of breast cancer [51].

Crosstalk with the elements of microenvironment inside the vessel has also huge impact on the EMT status of CTCs; tumor-activated platelets have been demonstrated to release $\alpha$-granules containing TGF-$\beta$ and ATP, promoting EMT in associated CTCs [52]. Indirect interactions with CAFs were also postulated in producing epithelial-mesenchymal plasticity [53].

### 4.3. Cell-Cell Junctions

CTC clusters retained at least partially epithelial characteristics, including high expression of cell–cell junction proteins. The most upregulated are: plakoglobin [35], keratin

14 [36], and claudin 11 [54]. Plakoglobin is a desmosomal or, less commonly, adherent junction protein and its expression was reported as an independent prognostic factor in breast cancer [55,56]. Keratin 14 is an intermediate filament protein associated with hemidesmosome; keratin 14 (+) cells were enriched in desmosomal and hemidesmosomal components [36]. Thus, it seems that desmosomes play an important role in CTC cluster formation. Claudins are internal membrane proteins and tight junction components, they form a physical barrier between cells, controlling the flow of solutes. Claudin upregulation in CTC clusters was demonstrated for squamous cell carcinomas [54]. Stem cell marker CD44, implicated in cluster formation, is also involved in the regulation of tight junctions [57]. Cell–cell junctions are crucial for cluster formation; knockdown of plakoglobin or CD44 precluded CTC cluster formation and suppressed metastasis [30,35].

In heterotypic clusters cell–cell junctions are also heterotypic; CAFs association with CTCs is mediated by E-cadherin-N-cadherin adhesions [58]. This direct interaction was shown to be crucial for directing clusters migration.

Interestingly, cell–cell junctions in clusters display remarkable plasticity and undergo dynamic re-modeling when clusters traverse narrow capillaries; it was shown that cells form a single file (without disintegrating), so they can fit into a vessel without blocking it [59].

## 5. Methods of Detection; Pros and Cons

The lack of the reliable CTC detection techniques represented for long decades a big stumbling block to CTC research. Last two decades were pivotal in this aspect and now we observe a multitude of effective techniques; however, there is still a problem with their standardization, compatibility, and reproducibility. Below, we present a short recapitulation of the main available techniques, with respect of their utility for CTC clusters detection.

There are several types of CTC isolation techniques, classified according to the principles of the method of their capture/detection: methods based on the enrichment or depletion of the CTC fraction achieved with specific antibodies (biological properties), isolation based on physical properties (size, density, cellular charge), microfluidics isolation which can use both size and specific markers and systems without prior selection, based on the detection in the whole sample. Mixed strategies are also often employed.

Enrichment with specific antibodies detecting proteins expressed on the surface of tumor cells is the most popular; the only (so far) FDA-approved method, CellSearch [60], falls into this category. However, CellSearch is not optimal for cluster analysis: the most obvious flaw is its dependence on EpCAM expression on tumor cell surface, which may not be the case for cells shifted toward mesenchymal characteristics. Second, in immunoaffinity-dependent systems, the enrichment step may not be as efficient for heterotypic clusters, covered with other cells or cell fragments (platelets), which restrict epitope accessibility.

The other immunoaffinity-based methods (AdnaTest [61], EasySep [62], Dynabeads [62], MINDEC- Multi-marker Immuno-magnetic Negative Depletion Enrichment of CTCs [63]) use mostly magnetic beads and employ either positive (e.g., EpCAM, cytokeratins) or negative (e.g., CD45) selection to enrich CTC fraction. These methods offer free selection of markers, but produce only enriched cellular fraction, thus, analyzing single CTCs or CTC clusters is impossible.

Filter-based methods (CellSieve [64,65], ISET [66], ScreenCell [67], FMSA [68]) should provide a good tool to study clusters, since clusters are much bigger than single cells of any type, but they also have their challenges; whole-blood filtration takes time and often leads to clogging, while using pressure to improve filtration may result in cell deformation or cluster disintegration.

The enrichment can be also achieved by density gradient centrifugation (Onco-Quick [69], Ficoll Paque [70]); these are simple centrifugation devices which enable to separate different types of blood cells. Naturally, these devices do not provide very specific enrichment, similarly to dielectric separation, based on differences in membrane area and charge of tumor cells versus blood cells (DEP). These methods can be used as tools to improve specificity of some other methods at the beginning or the end of other procedures.

The best promise, especially with respect to CTC clusters is probably held by microfluidic devices, often coupled with some other method. Microfluidic devices used in CTC isolation are numerous and, in essence, are based on different size or deformability of CTCs, which implies their different flow pattern. They can rely only on physical properties, which has an advantage of being label-free and hence, independent of the specific markers (Parsortix [71], Labyrinth [72], JETTA [73]). Some of them were specially designed to isolate CTC clusters (deterministic lateral displacement: DLD [74], Straight Microfluidic Chips [75]). Another type of microfluidic devices combines specific flow pattern with immunoaffinity, boosting specificity, but—as with any immunoaffinity-based method—limiting sensitivity (CTC-iChip [76], CMx platform [77], HB-CTC-Chip [78,79], OncoCEE [80]). Microfluidic approaches are numerous, from very basic to extremely complicated, many of them are custom-made, but in the selection process some may achieve the status of clinically approved method (for example, Parsortix is currently waiting for FDA approval). They offer high specificity, are relatively gentle and enable isolation of viable cells for further analysis. They seem to hold a big promise for the efficient and reliable CTC isolation, especially regarding CTC clusters.

Different type of methods is represented by imaging-based approaches, working without any specific enrichment apart from basic centrifugation, which produces buffy coat layer containing nucleated blood cells and CTCs (RareCyte: [81], CytoTrack [82]). Material from this layer is stained with the appropriate antibodies, spread on a disk or a slide, and examined microscopically. In the CytoTrack system single cells can be picked from the disk using micromanipulator (Cytopicker) and their genetic material can be examined, but since they are fixed, no other types of analysis are available. This approach is free from the flaws of immunoaffinity-dependent enrichment and can be as specific as are the antibodies used to detect CTCs.

Summary of the CTC detection methods described here is presented in Figure 3 and Table 2.

**Table 2.** Description of CTC detection methods presented in Figure 2 with the assessment of their usefulness in CTC cluster analysis. For more detailed description see review [83].

| Category | Methods (Examples) | Key Features | Ref. |
|---|---|---|---|
| Biological properties | CellSearch, AdnaTest, EasySep, Dynabeads, MINDEC | Immunoaffinity-based (marker-dependent isolation), sensitivity in a range from 27% (CellSearch) up to 70% (CellSearch)–73% (AdnaTest), specificity 89–99% (CellSearch), clinically validated (CellSearch), EpCAM(−) cells may be lost, not optimal for cluster isolation | [60–63] |
| | RareCyto, CytoTrack | Imaging-based approaches (marker-dependent identification), detection of EpCAM(+) and EpCAM(−) cells, clusters can be observed (Figure 1) | [81,82] |
| Physical properties | OncoQuick, Ficoll Paque | Density-gradient centrifugation, marker-independent, low purity of CTCs | [69,70] |
| | Parsortix, Labyrinth, JETTA, DLD, Straight Microfluidic Chips | Mircrofluidics (size and deformity based, marker-independent isolation), high sensitivity (92%, Parsortix) and specificity (100%, Parsortix), subsequent analysis of cells possible, suitable for CTC cluster analysis, some specially designed for this purpose (DLD, Straight Microfluidic Chips) | [71–75] |
| | CellSieve, ISET, ScreenCell, FMSA | Filter-based methods, marker-independent, some allow post-capture culture and microscopic examination (ScreenCell), sensitivity 76%, specificity 82% (ISET), size-dependent methods favor cluster isolation, but applied pressure may disrupt clusters | [64–68] |
| Mixed properties | CTC-iChip, OncoCEE, Cmx platform, HB-CTC-Chip | Immunoaffinity with Microfluidics, high sensitivity (95%, OncoCEE) and specificity (92%, OncoCEE), suitable for CTC cluster analysis | [76–80] |

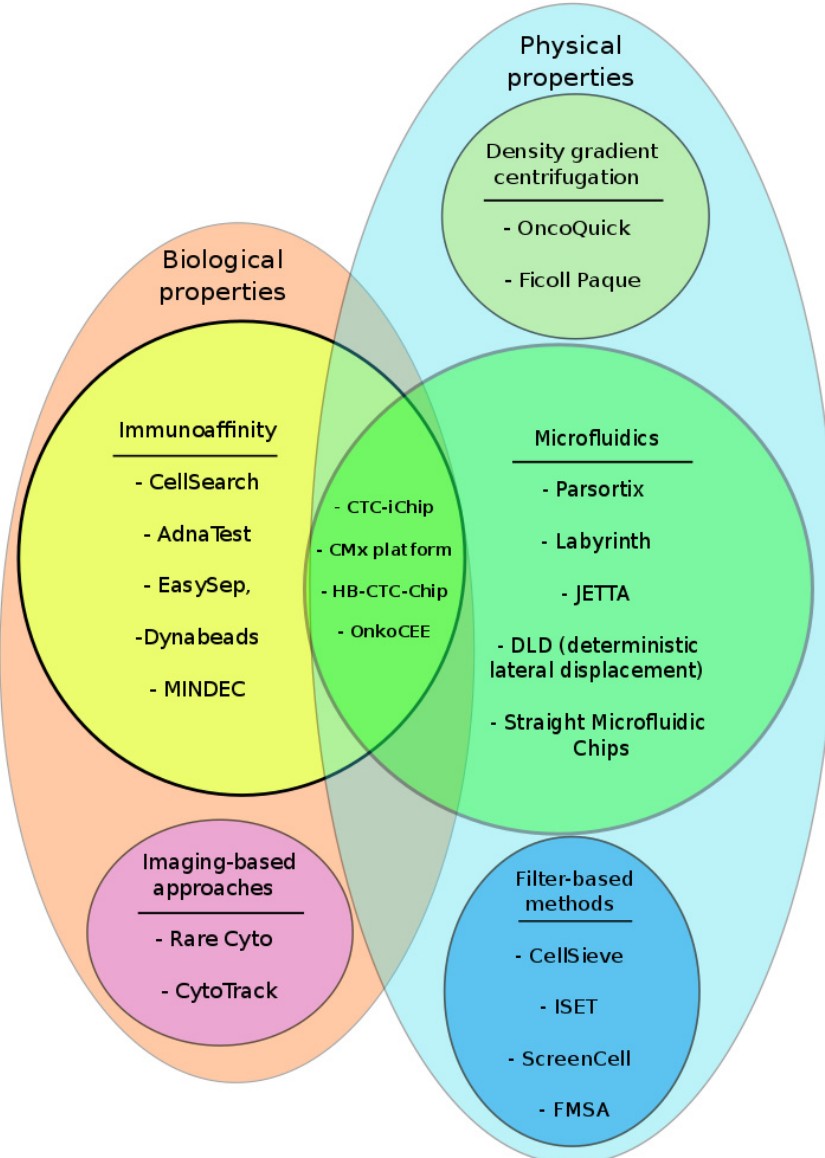

**Figure 3.** Methods commonly used in CTC isolation. Methods classified according to biological and physical properties, with some using mixed techniques, to improve specificity (overlapping ellipses).

## 6. CTC Clusters and Their Impact on Survival

Single CTCs are well-known prognostic factors in breast cancer patients. The identification of one or more CTCs in 7.5 mL of blood correlates with early recurrence and decreased overall survival (OS) in chemonaive patients with stage I-III [84]. In metastatic breast cancer patients the presence of CTCs is associated with increased risk of progression and shorter OS [85]. The predictive value of CTCs detection remains unclear and it is not recommended and routinely used in a clinical practice. The data coming from small, prospective study confirm that single CTCs and CTC clusters enumeration can be useful in advanced breast cancer patients because of its effect on PFS [86]. Moreover, increased number of CTC clusters between day 15 and 29 of a designed study cycle was related to poorer PFS in metastatic triple negative breast cancer patients [87]. Respectively, patients with MBC and prolonged presence of CTC clusters in blood are characterized by worse prognosis [35]. Considering the fact that majority of single CTCs die in a bloodstream, further research on CTC clusters and their impact on metastatic process is needed. CTC clusters were found as more potent and resistant to apoptosis in comparison to single CTCs in a mouse model in reference to the formation metastatic lesions in lungs [35].

Recognition of clusters as potential therapeutic targets opens new fascinating treatment possibilities. Studies on the methylation status of CTC clusters lead to a conclusion that their disruption may be beneficial for the patient. Gkountela et al. [44] describe identification of the compounds that enable dispersion of CTC clusters into single cells, suppressing metastasis formation, due to lower metastatic potential of single cells. They suggest that these compounds should be administered early on, at the time of localized disease, to prevent the spread, or at later stages to prevent metastasis-to-metastasis seeding. These compounds represent the FDA-approved cardiac glycosides ouabain and digitoxin. They function as Na+/K+ ATPase inhibitors, leading to the increase of intracellular Ca2+ concentration and subsequent disruption of desmosomal and tight junctions and cluster disintegration.

Another surprising possibility was described by Donato et al. [38]. The authors observed that hypoxia leads to cell–cell junction upregulation, followed by the formation and intravasation of highly metastatic CTC clusters. As a consequence, they have come to rather disturbing realization that although anti-angiogenic treatment leads to primary tumor shrinkage, it also increases intra-tumor hypoxia, leading to the increased cluster shedding. Conversely, while pro-angiogenic treatment results in primary tumor enlargement, it also suppresses the formation of CTC clusters, limiting metastasis.

## 7. Conclusions

CTC clusters appear to be the main factor and the most important target in breast cancer metastasis. While they are obviously extremely rare, the current methods of CTC isolation may underestimate their content for technical reasons. With the focus shifted to CTC clusters, new experimental and treatment approaches can be developed, which can be especially useful in early stages to prevent dissemination, or even in late stages, to prevent metastasis-to-metastasis seeding, observed in breast cancer patients.

**Funding:** This research was funded by Polish National Science Centre, grant numbers 2016/21/B/NZ2/03473, 2019/33/N/NZ5/00758.

**Institutional Review Board Statement:** The study (cell imaging for Figure 1) was conducted according to the guidelines of the Declaration of Helsinki, and approved by the Institutional Ethics Committee of National Institute of Oncology (protocol code 34/2016, 9 September 2016).

**Informed Consent Statement:** Informed consent was obtained from all subjects involved in the study.

**Data Availability Statement:** Not applicable.

**Conflicts of Interest:** The authors declare no conflict of interest.

## Abbreviations

| | |
|---|---|
| CAF | cancer associated fibroblast |
| CK | cytokeratins |
| CTC | circulating tumor cell |
| DEP | dielectrophoresis |
| ECM | extracellular matrix |
| EpCAM | Epithelial Cell Adhesion Molecule |
| EMP | epithelial-mesenchymal plasticity |
| EMT | epithelial-mesenchymal transition |
| HER2 | human epidermal growth factor receptor |
| HR | hormone receptors |
| MBC | metastatic breast cancer |
| NET | neutrophil extracellular traps |
| TAN | tumor-associated neutrophils |
| TME | tumor microenvironment |
| TNBC | triple negative breast cancer |
| WBC | white blood cells |

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
