# Peer review of "Clusters, Assemblies and Aggregates of Tumor Cells in the Blood of Breast Cancer Patients; Composition, Mode of Action, Detection and Impact on Metastasis and Survival"

_2673-8937, doi:10.3390/ijtm1010005_

Round 1

Reviewer 1 Report

This a complete review on composition and functional roles of circulating tumor cell clusters in breast cancer. Authors describe and discuss the types of CTC clusters and cellular assemblies present in the circulation, the origin of CTC clusters, the markers and phenotypes of CTC clusters, their methods of detection and prognostic meaning.

Minor points:

Some abbreviation are lacking. Line 304: ”differences in membrane area and charge of tumor cells versus blood cells (DEP)”. Specify: (Dielectrophoresis, DEP) 

Greek letters are lacking in the text. See lines 83 and 245.

Author Response

Minor points:

Some abbreviation are lacking. Line 304: ”differences in membrane area and charge of tumor cells versus blood cells (DEP)”. Specify: (Dielectrophoresis, DEP)

Abbreviation is added.

Greek letters are lacking in the text. See lines 83 and 245.

Greek letters are corrected.

Reviewer 2 Report

The authors surveyed methods of Circulating tumor cells (CTCs) detection and their role in metastasis and survival, and new approaches to therapy. The paper sound interesting and well organized, However, I have minor concerns:

  • The authors may highlight more in the merits and demerits of each type of method.
  • The manuscript would be better if there is a numeric comparison between the methods (Accuracy, Precision, Sensitivity, Specificity).
  • reference number 66 in the list of references is capitalized ( the authors) which is inconsistent with the other references.

Author Response

The authors may highlight more in the merits and demerits of each type of method.

The manuscript would be better if there is a numeric comparison between the methods (Accuracy, Precision, Sensitivity, Specificity).

To comply to these comments a table was added (Table 2), which specifies key features of a described method, sensitivity and specificity (in methods to which these parameters were assigned) and their usefulness in regard to CTC clusters.

reference number 66 in the list of references is capitalized ( the authors) which is inconsistent with the other references.

The reference was corrected.

Reviewer 3 Report

The authors provide nice introduction and techniques currently used in circulating tumor cells (CTCs) detections. The review overall is informative. Some of the comments are as follows:

The introduction part heavily writes on different types of breast cancer. It would certainly be better to focus on discovery and usage of CTC.

Lines 53-59 should include necessary citations.

In figure 2, the authors could include a table for the markers used for detection of the types of CTC assemblies.

Similarly Figure 3, will be better if a table, even separately, included with references to the techniques shown in the Venn diagram. Is there any comparison literature available for relative merits and de-merits of these? Those types of references can be includes in a table form.

Author Response

The introduction part heavily writes on different types of breast cancer. It would certainly be better to focus on discovery and usage of CTC.

The introduction was re-written, a description of breast cancer types was shortened and a part pertaining to the discovery and usage of CTC was expanded.

Lines 53-59 should include necessary citations.

Citations were added.

In figure 2, the authors could include a table for the markers used for detection of the types of CTC assemblies.

A table with markers used for detection was added (Table 1).

Similarly Figure 3, will be better if a table, even separately, included with references to the techniques shown in the Venn diagram. Is there any comparison literature available for relative merits and de-merits of these? Those types of references can be includes in a table form.

A table summarizing key features of a described methods, along with their usefulness for CTC cluster analysis and a reference to the appropriate review was added (Table 2).